# Confinement of Masonry Columns with the FRCM-System: Theoretical and Experimental Investigation †

**Maria Antonietta Aiello [1,2], Alessio Cascardi [3], Luciano Ombres [4,*] and Salvatore Verre [4]**

[1] Department of Innovation Engineering, University of Salento, 73100 Lecce, Italy; antonietta.aiello@unisalento.it

[2] ITC-Construction Technologies Institute, CNR-Italian National Research Council, San Giuliano Milanese, 20098 Milan, Italy

[3] ITC-Construction Technologies Institute, CNR-Italian National Research Council, 70124 Bari, Italy; alessio.cascardi@itc.cnr.it

[4] Department of Civil Engineering, University of Calabria, 87036 Cosenza, Italy; salvatore.verre@unical.it

[*] Correspondence: luciano.ombres@unical.it

† OPEN ISSUE FOR CONFINEMENT OF MASONRY COLUMNS WITH FRCM-SYSTEM: THEORETICAL AND EXPERIMENTAL INVESTIGATION accepted Rehabend conference 2020.

**Abstract:** *Fabric Reinforced Cementitious Matrix* (FRCM) systems are promising solutions for the confinement of masonry columns because they demonstrate strengthening effectiveness and, at the same time, compatibility with historical substrates. Nevertheless, the matrix is responsible for the stress-transfer from the structural element to the fabric-reinforcement. Therefore, in the case of poor-quality mortar, the effectiveness of the strengthening can be limited or even compromised. On the other hand, the low content of fibers utilized for FRCM systems generally involves the need to apply more layers in order to accomplish design requirements and a continuous configuration of the reinforcement is more often addressed. Few experimental and theoretical investigations have been targeted to the before mentioned aspects in the recent past, namely the influence of the kind of mortar, the number of layers, and the strengthening configuration (continuous, discontinuous) on the effectiveness of confinement. The present paper refers to the results of an experimental investigation on FRCM confined clay brick masonry. A series of small-scale masonry columns were tested under monotonic centered load until collapse. The varied parameters were the number of confining layers (i.e., 1, 2, and 3) and the confinement configuration (i.e., continuous and discontinuous). The performed research aims to contribute in strengthening to the knowledge in the field of FRCM-confinement, mainly focusing on some of the mentioned unexplored aspects (number of layers, strengthening configuration) that could be considered for validation/improvement of analytical design-oriented formulas. In particular, some analytical models, available in the technical literature, were adopted for predicting the herein reported experimental results. Even if based on few experimental results, the outcomes showed that the number of FRCM-layers and the confinement configuration were crucial parameters affecting the confining effectiveness. The compressive strength was satisfactorily predicted in all cases by the two available utilized models. On the other hand, an improvement in the utilized AOM model is suggested in order to include the stress–strain curves of the hardening type.

**Keywords:** masonry; confinement; PBO-FRCM; testing; theoretical model

## 1. Introduction

The past use of fiber reinforced polymers (FRPs) for the confinement of masonry columns evidenced some critical aspects such as the difficult reversibility and the poor compatibility with the existing substrate. If the matter related to the reversibility could be solved [1–3], the problem of compatibility remains an open issue related to the nature of the FRP-matrix (generally epoxy-based). For this reason, the combination between inorganic matrix and fibers is considered as an advantageous alternative; several investigations have been carried out in the last decade, demonstrating the structural benefit of the strengthening [4–11]. These new families of composites are known as a fabric reinforced cementitious matrix (FRCM) or textile reinforced mortar (TRM). Inorganic matrices are used with carbon, PBO (polyparaphenylenebenzobisoxazole) [12], basalt [13], glass [14–16], carbon [17], and steel fibers, generally in a mesh configuration. When unidirectional steel cords are combined with mortar, the composites are better known as steel reinforced grout (SRG) [18,19]. The behavior of FRCM/SRG confined masonry columns has been extensively investigated mainly by experimental research. Tests have been conducted on columns under both concentric and eccentric loads with varying geometrical and mechanical properties. The main parameters investigated were the confinement ratio (i.e., the number of confining plies) [19–22], the mortar grade [23], the corner radius [24], and the load eccentricity [25]. Nonetheless, the theoretical prediction of the axial strength of the FRCM/SRG-confined column is difficult to assess. In fact, the parameters related not only to the properties of the fiber and of the substrate, but also to the properties of the matrix and its interaction with the fabric should be taken into account. The available analytical models need to be deeply validated, especially for the cases of multi-layer confinement and discontinuous confinement [26–29]. For this reason, the present study focused on the experimental investigation including the multi-layer reinforcement and discontinuous configuration as well as the comparison between the experimental results and theoretical predictions on the basis of available models.

## 2. Experimental Work

### 2.1. Specimen Details

The experimental campaign, herein reported, investigated a strengthening solution for masonry heritage buildings throughout the confinement of a series of masonry columns by a PBO-FRCM system (PBO fiber and cement-based matrix). The aim was to evaluate the structural performances in terms of axial strength and ductility. In addition, the effects of the n-layer and the arrangement of confinement, continuous and discontinuous, were both analyzed. The layout of the experimental program is reported in Table 1, in which the radius of curvature of the corners ($r_c$), the thickness of the FRCM-matrix ($t_{mat}$), the equivalent thickness of the fabric ($t_{feq}$), the number of confining layers (n), the confinement scheme, and the ratio between the width ($w_f$) and the spacing ($s_f$) of the confining jackets are detailed per specimen. The experimental set, herein reported, is part of a larger on-going experimental program. Seven small-scale columns were considered of which one column was used as the control specimen. Three columns were confined with a continuous strengthening system by one, two, and three layers, respectively, while the remaining three columns were strengthened with a discontinuous configuration, by one, two, and three layers. Figure 1 shows the geometry of the unconfined specimen and the columns reinforced by a continuous and a discontinuous system.

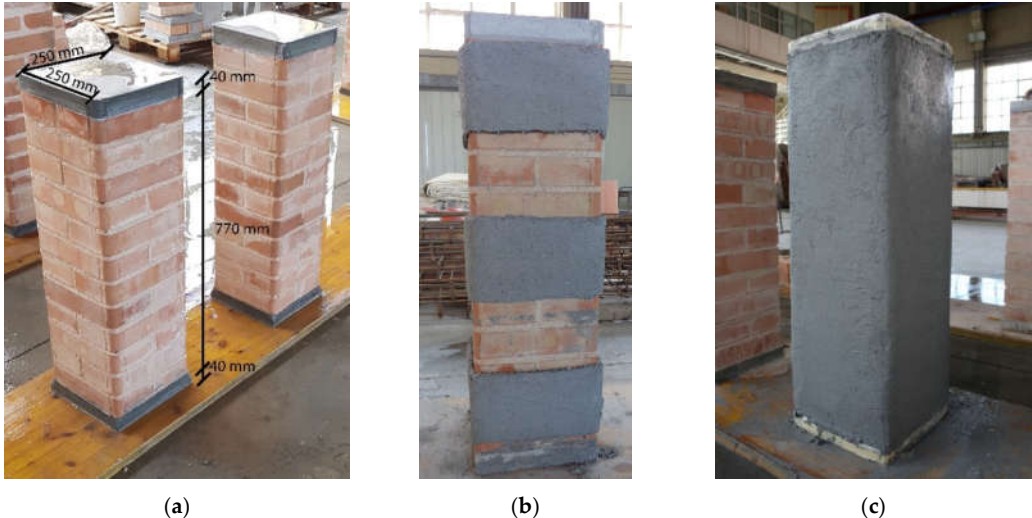

|  |  |  |
|:--:|:--:|:--:|
| (**a**) | (**b**) | (**c**) |

**Figure 1.** Small scale column: (**a**) geometry, (**b**) discontinuous confinement, and (**c**) continuous confinement.

The specimens had a square cross-section equal to 250 × 250 mm² and 770 mm in height. The confined specimens were named following the designation C-X-P-nL, where C indicates the confined specimen, X indicates the type of confining configuration (C or D for continuous or discontinuous, respectively), P the type of fiber used (PBO in this experimental campaign), and nL indicates the number of layers. Referring to the discontinuous configuration, the utilized strips were spaced 300 mm (distance between their middle lines) while the width of each strip was 150 mm.

**Table 1.** Specimen descriptions.

| Label | $t_{mat}$ (mm) | n (-) | Confinement Scheme | $w_f/s_f$ |
|:--:|:--:|:--:|:--:|:--:|
| UCS | - | - | - | - |
| C-C-P-1L | 6 | 1 | Continuous | 1 |
| C-C-P-2L | 9 | 2 | Continuous | 1 |
| C-C-P-3L | 12 | 3 | Continuous | 1 |
| C-D-P-1L | 6 | 1 | Discontinuous | 0.5 |
| C-D-P-2L | 9 | 2 | Discontinuous | 0.5 |
| C-D-P-3L | 12 | 3 | Discontinuous | 0.5 |

Moreover, to avoid a premature compression failure in the masonry and, at the same time, distribute the applied load uniformly, a 40 mm thick layer of high-strength mortar was realized on the top and bottom of the columns.

### 2.2. Strengthening System, Specimen Preparation, and Test Setup

The strengthening system was made of a PBO unbalanced fabric mesh embedded in a cement-based matrix; in the principal direction (longitudinal), it had the equivalent thickness, $t_{feq}$, equal to 0.046 mm, while in the transversal direction, the $t_{feq}$ was equal to 0.012 mm, according to the manufacturer technical sheet [30]. Mechanical properties of the dry fabric were determined in the longitudinal direction by tensile tests [31] according to the test setup in Figure 2a. The tests were conducted in displacement control at 0.5 mm/min and, in order to evaluate the local strain, an extensometer was placed at the middle of the dry textile. The average values of the elastic modulus, tensile strength, and ultimate strain ($\varepsilon_{fu}$) were 211.4 ± 8.67 GPa, 3.40 ± 0.10 GPa, and 0.025 ± 0.002 (mm/mm), respectively. Direct tensile tests were also carried in order to determine the elastic modulus, tensile strength, and ultimate strain of the FRCM system. Five prismatic coupons (FRCM specimens) with a nominal size of 500 × 50 × 10 mm³ were adopted. The dry textile was placed at the

middle of the coupon with the longitudinal yarns in the loading direction. The fabric meshes of PBO were composed of five yarns. Two aluminum tabs were attached to the ends of each specimen for a length of 100 mm, in order to assure a good stress distribution during the test. Figure 2b reports the test setup adopted.

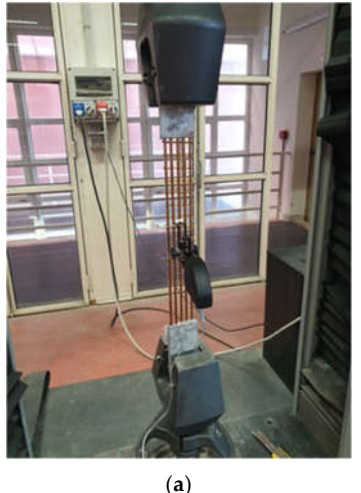 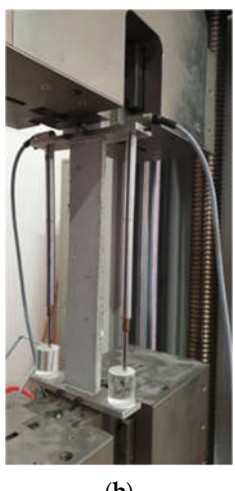

(**a**)　　　　　　　　　　　　　　　　　　(**b**)

**Figure 2.** Test setup: (**a**) dry fabric and (**b**) FRCM specimen.

Tests were conducted in the displacement control and the local strain was measured by two LVDTs (linear variable displacement transducers) placed on the gauge length. The average values of the elastic modulus at the cracked stage, tensile strength, and ultimate strain obtained by the tests were 94 ± 8.46 GPa, 1598 ± 272 MPa, and 0.0125 ± 0.0003 (mm/mm), respectively. Compression and flexural tests were conducted to determine the mechanical properties of the pozzolanic mortar (see Figure 3), according to UNI EN 12190:2000 [32] and UNI EN 1015-11 [33]. The average values of the compressive strength and the flexural strength were 36.16 ± 1.45 MPa and 5.50 ± 0.77 MPa, respectively. Figure 3 reports the test setup adopted.

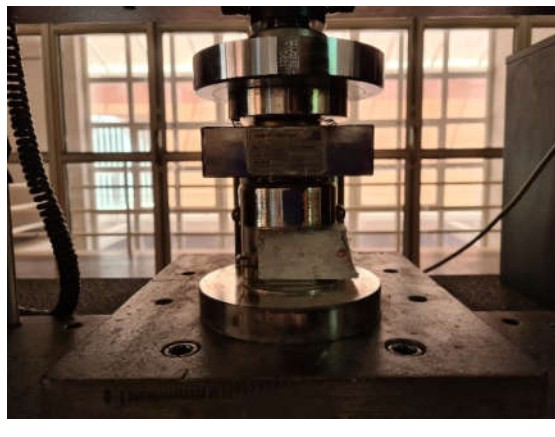 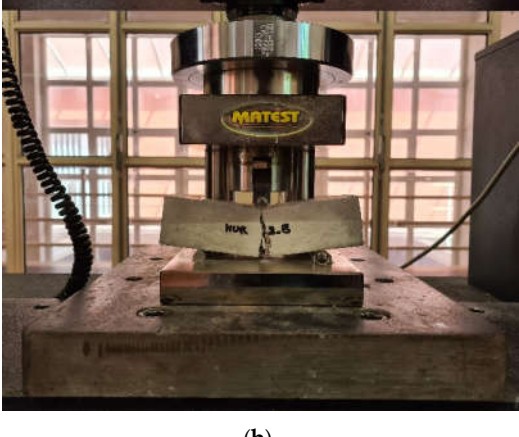

(**a**)　　　　　　　　　　　　　　　　　　(**b**)

**Figure 3.** Test setup: (**a**) compression and (**b**) flexural test on the matrix.

The confined columns were realized with a round corner of 20 mm that was wetted before installing the composite jacket. The composite jacket was applied starting at a distance of 10 mm from the top and the bottom of the column (see Figure 4). It is possible to divide the application of the strengthening system in three different steps. First, the internal mortar layer is applied as well as auxiliary strips of polystyrene aimed to control the whole thickness of the reinforcement (see Figure 4a); then (second step), a single layer of sheet/fabric mesh was disposed on the internal mortar layer,

pushing it into the matrix carefully in order to guarantee an appropriate penetration (see Figure 4b) and subsequent wrapping around the column. Finally (third step), the external mortar layer was applied (see Figure 4c). Moreover, the multi-ply confinement was achieved by repeating steps 2 and 3.

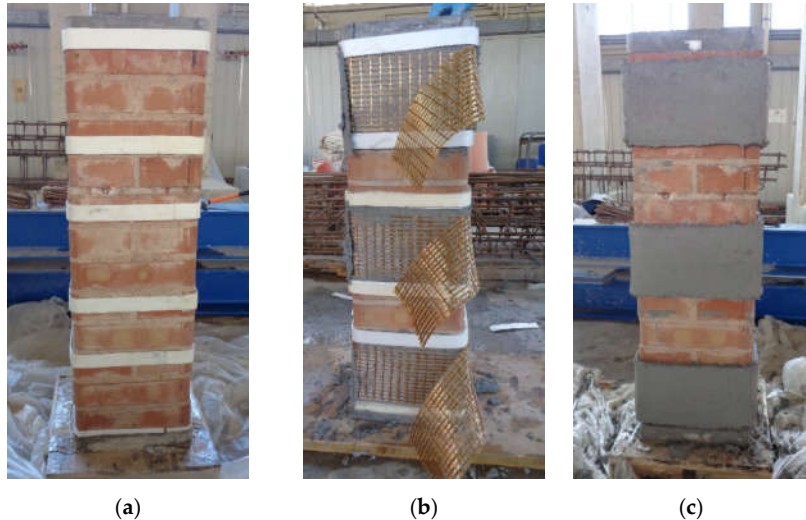

(**a**)          (**b**)          (**c**)

**Figure 4.** Confinement steps: (**a**) application of the internal mortar layer, (**b**) fabric mesh installation, and (**c**) application of the external mortar layer.

The thickness of each mortar layer was assumed equal to 3 mm, as recommended by the manufacturer's instructions. In the case of multi-ply confinement, the overlapping lengths were properly set on a different cross-section side per ply. In such a way, a weak confined side was avoided as well as the potential formation of a vertical fabric-opening near the column's corners (see Figure 5). After casting, the masonry columns were cured in the laboratory under wet cloths for 28 days prior to testing. Figure 5 also shows the loading system. In detail, the load was applied by a hydraulic jack installed at the top of the column. Each test was conducted by the loading control method with a load rate of approximately 40 N/s. It should be noted that at the top and the bottom of the column, a steel plate with a thickness of 20 mm was placed to guarantee a uniform distribution of the compression load. The test was considered ended when a significant drop in the load was registered after the peak load.

The vertical and lateral displacement were measured by four and twelve LVDTs, respectively (Figure 5). In particular, the vertical LVDTs were installed at the corners of the columns. A total of twelve LVDTs were positioned along the height of the columns (i.e., three LVDTs for each face of the column), in particular at the top, at the mid height, and at the bottom.

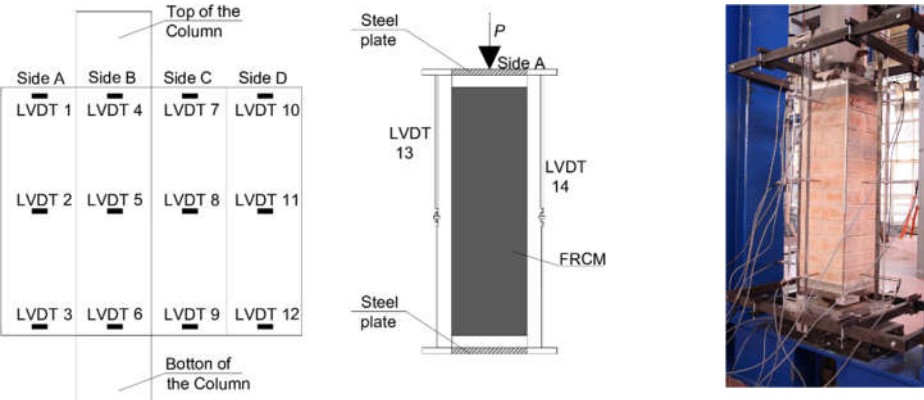

**Figure 5.** Test setup adopted.

## 2.3. Experimental Results and Discussion

Different failure modes were observed for the tested masonry specimens. For the un-confined column, the failure was brittle due to the crushing of the masonry. The failure configuration was characterized by the presence of a vertical crack, visible at the middle side of the specimens (see Figure 6). The failure of the continuous confined specimens occurred in all cases with the formation of a wide vertical crack, mainly in correspondence with two corners of the columns, causing the rupture of the PBO fibers while the masonry resulted in being extensively damaged. The failure mode observed in the case of the discontinuous confined columns was different from that of the continuously confined ones. During the test, a vertical crack was observed at the middle side of the column (Figure 6d). Then, other cracks appeared on the external reinforcement, particularly at the overlapping zone, while the unconfined part of the column was clearly damaged. Immediately before the collapse, the external reinforcement at the middle of the column (overlapped zone) opened completely.

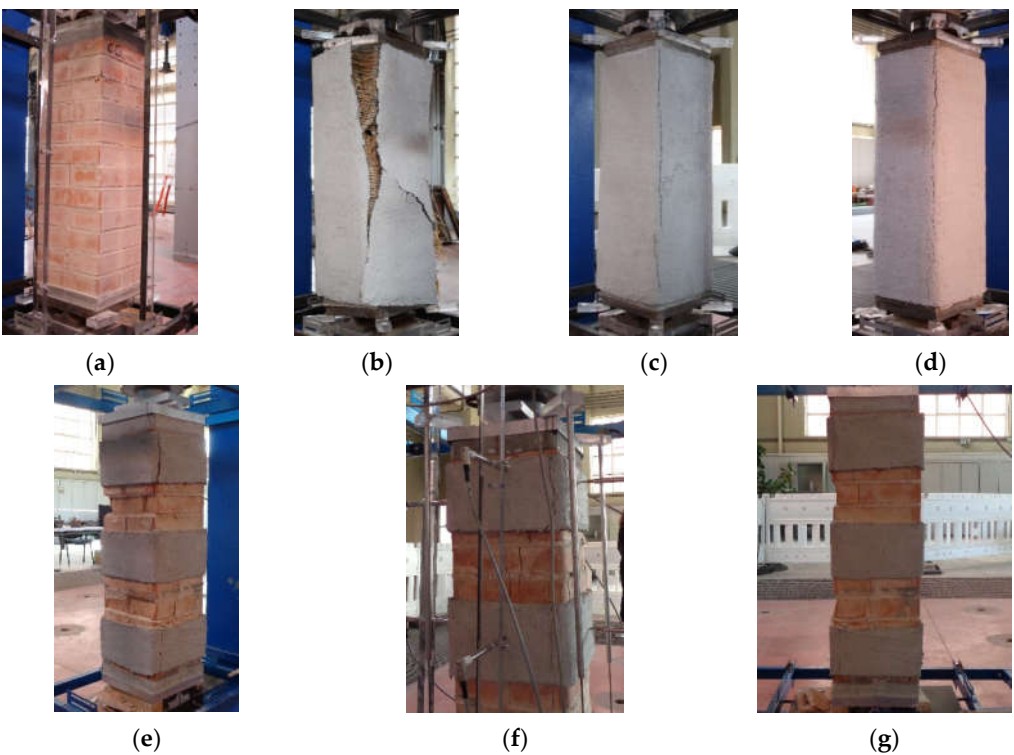

**Figure 6.** Failure configurations of confined columns: (**a**) un-confined column, (**b**) C-C-P-1L, (**c**) C-C-P-2L, (**d**) C-C-P-3L, (**e**) C-D-P-1L, (**f**) C-D-P-2L, and (**g**) C-D-P-3L.

### 2.3.1. Axial Stress–Strain Law

Figure 7 shows the stress–strain curves for both the unconfined and confined columns, while in Table 2, the values of elastic modulus ($E_{el}$), the peak axial stress ($\sigma_P$) and the corresponding axial strain ($\varepsilon_P$), the ultimate lateral strain ($\varepsilon_{lu}$), the confinement ratio ($\xi$), and the ductility ($\mu$) are reported for all specimens. In particular, the confinement ratio, $\xi$, is defined as the ratio between the axial strength of the confined specimen and that of the unconfined one. The axial stress–strain curves can be approximated with a bi-linear trend. The first linear part has a slope similar to that of the un-confined masonry while the second linear part is of the hardening type with a slope affected by the number of FRCM-ply.

For the continuously confined columns, the gain of the axial strength increased with the number of FRCM-plies while ultimate strain resulted in being quite similar. Similar consideration can be made in the case of discontinuous confined columns even if the strengthening effectiveness was lower, as expected, due to the lower amount of confined volume of masonry. The ductility of the confined columns was computed as the ratio between the difference of the ultimate strain and the elastic strain over the elastic strain. It should be noted that the ductility was not influenced by the number of plies. On the other hand, the influence of the confining scheme was clear since the ductility increased up to an average value of 345% and 176% for continuous and discontinuous FRCM-wrapping, according to Table 2. In both cases, the ductility of the unconfined masonry (i.e., 84%) was largely increased.

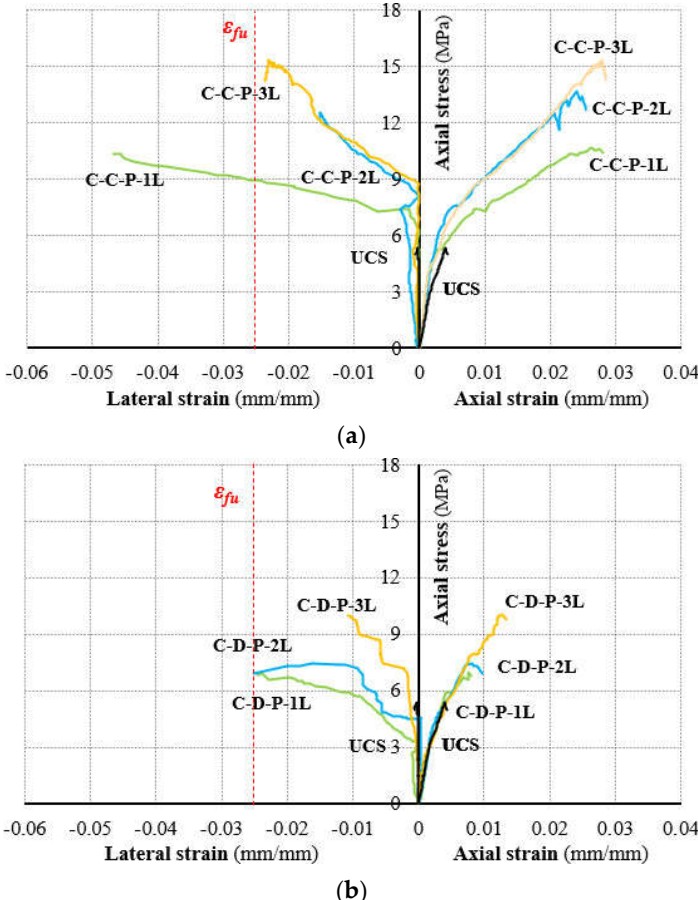

**Figure 7.** Experimental axial stress–strain curves: (**a**) continuous and (**b**) discontinuous confined columns.

In the case of the discontinuous configuration, the axial strength was lower than that recorded for the full-wrapping (single layer); this result, as expected, was due to the significant damage of the unconfined parts of the columns. The elastic modulus was calculated by evaluating the secant slope of the axial stress–strain curve between 5% and 40% of the peak axial stress (i.e., elastic branch). From the values in Table 2, it is possible to observe that the maximum initial stiffness was obtained for the column confined with 1-layer, while it decreased with an increase in the number of FRCM-plies. Furthermore, the scatter in terms of elastic modulus was not very significant; this aspect needs deeper investigation since only one specimen per series was tested in the present experimental program.

**Table 2.** Test results.

| Label | $E_{el}$ (MPa) | $\sigma_p$ (MPa) | $\varepsilon_p$ (mm/mm) | $\varepsilon_{lu}$ (mm/mm) | $\xi$ (-) | $\mu$ (-) | $\beta = \varepsilon_{lu}/\varepsilon_{fu}$ (-) |
|---|---|---|---|---|---|---|---|
| UCS | 1510 | 5.19 | 0.0025 | 0.003 | - | 0.84 | - |
| C-C-P-1L | 3181 | 10.59 | 0.0026 | 0.047 | 2.04 | 3.38 | 1.88 |
| C-C-P-2L | 2608 | 13.60 | 0.0259 | 0.015 | 2.62 | 3.32 | 0.60 |
| C-C-P-3L | 3184 | 15.23 | 0.0279 | 0.023 | 2.93 | 3.65 | 0.92 |
| C-D-P-1L | 2321 | 6.97 | 0.0077 | 0.024 | 1.34 | 1.56 | 0.96 |
| C-D-P-2L | 1909 | 7.41 | 0.0081 | 0.025 | 1.42 | 1.70 | 1.00 |
| C-D-P-3L | 1443 | 10.02 | 0.0128 | 0.011 | 1.93 | 2.10 | 0.44 |

A β-parameter was also computed and reported in Table 2 according to the ratio between the maximum lateral strain and the ultimate strain of the dry fabric in the tensile stress-state (see Section 2.2). In such a way, the potential confining action of the FRCM-jacket is intended to be evaluated. In

fact, β = 1 means that the FRCM-jacket reached the maximum FRCM deformation capacity. It can be noted that in almost all cases, the β-value was close to 1 (i.e., C-C-P-3L, C-D-P-1L, and C-D-P-2L). Only C-C-P-1L manifested a value over than 1. This appears physically senseless because the maximum deformation of the PBO-fabric was overcome. In reality, the FRCM-jacket demonstrated a failure dominated by the jacket opening itself. Thus, the recorded lateral deformations were significantly increased. In addition, as evident from Figure 7, the lateral strain recordings of C-C-P-1L mainly started when a crack opened within the FRCM-matrix at axial stress levels more than 6 Mpa, while a linear trend from the origin of the graph was expected when the recordings were related to the dilatations. Furthermore, in the case of specimens C-C-P-2L and C-D-P-3L, the lateral strain capacity was significantly lower than the $\varepsilon_{fu}$; therefore, the potential FRCM-confining effectiveness did not seem to be used at all. Further tests concerning this series may overcome this result since a single specimen per series may be a limit for the repeatability of the phenomena.

The herein obtained results show evidence that the higher the confinement ratio, the higher the compressive strength of the confined columns for both full and partial confinement schemes even if with different post peak trends. In fact, when the ξ-value passed from 2.04 to 2.93 in the case of full FRCM-wrapping, the post-peak slope increased. In the case of partial confinement, the different confining ratio produced significant differences in terms of strength while the post-peak slope remained similar. This last result mainly appears to be related to the damage of the unconfined zone of the columns that, in the last stage, dominated the mechanical response.

## 3. Theoretical Prediction

The axial response of a FRCM-confined masonry column is affected by cracks opening both into the masonry core and in the jacket. For this reason, the analytical prediction of the axial strength cannot disregard the parameter related to the masonry, the fabric, and the FRCM-matrix. According to this consideration, two available analytical models were implemented herein for comparison with the experimental data [26,28]. In 2007, Cascardi et al. [26] developed a design formula by empirically fitting the literature experimental data through the multiple linear regression method. In 2018, the Italian Guidelines concerning FRCM-strengthening of existing structures was published by CNR (National Research Council) [29] included a chapter concerning "*confinement*".

The comparison of the two proposals is reported in Table 3 according to Equations (1)–(8). The main difference is in computing the coefficient of efficiency due to the adopted FRCM-matrix, $k_{mat}$ (Equation (4)). Moreover, the relationship between this last parameter and the predicted axial strength of the confined column is unlike (Equation (8)): a linear proportion was used in [26], while a root square was proposed by [29]. In addition, the CNR-code considers a further parameter (i.e., the mass density of the masonry $g_m$) (Equation (7)).

**Table 3.** Description of the analytical models.

| Equation | Variable | Description | [26]-Cascardi et al. (2017) | [29]-CNR DT 215 (2018) |
|:---:|:---:|:---:|:---:|:---:|
| (1) | $k_H$ | Horizontal geometrical efficiency coefficient | $1 - \dfrac{(b - 2r_c)^2 + (h - 2r_v)^2}{3bh}$ | |
| (2) | $k_V$ | Vertical geometrical efficiency coefficient | $\left(1 - \dfrac{\rho_f}{2min(b;h)}\right)^2$ | |
| (3) | $\rho_{mat}$ | Geometrical percentage of FRCM-matrix | $4\dfrac{t_{mat}}{f_m}$ | |
| (4) | $k_{mat}$ | FRCM-matrix efficiency coefficient | $6\left[\rho_{mat}\left(\dfrac{f_{c,mat}}{f_m}\right)\right]^1$ | $1.81\left[\rho_{mat}\left(\dfrac{f_{c,mat}}{f_m}\right)\right]^2$ |
| (5) | $f_l$ | Confining pressure | $\dfrac{2nt_{feq}E_f\varepsilon_f}{D}$ | |
| (6) | $f_{l,eff}$ | Effective confining pressure | $k_H k_V f_l$ | $k_{mat} k_H k_V f_l$ |
| (7) | $k'$ | Masonry typology efficiency coefficient | - | $\dfrac{g_m}{1000}$ |
| (8) | $f_{cm}$ | Compressive strength of the FRCM-confined column | $f_m\left[1 + k_{mat}\left(\dfrac{f_{l,eff}}{f_m}\right)^{0.5}\right]$ | $f_m\left[1 + k'\left(\dfrac{f_{l,eff}}{f_m}\right)^{0.5}\right]$ |

where

- *b* is the width of the column cross-section (mm);
- *D* is the diagonal of the column cross-section (mm);
- $E_f$ is the Young modulus of the fabric (MPa);
- $f_{c,mat}$ is the compressive strength of the FRCM-matrix (MPa);
- $f_m$ is the compressive strength of the un-confined column (MPa);
- $g_m$ is the mass density of the masonry (kg/m$^3$);
- *h* is the height of the column cross-section (mm);
- $\rho_f$ is the spacing in between two consecutive strips (mm); and
- $\varepsilon_f$ is the ultimate tensile strain of the fabric (-).

Using the analytical models, the mean ± Co.V (coefficient of variation) and the median of the ratio between the experimental and theoretical value of the axial strength of the FRCM-confined column, $\frac{f_{cm,exp}}{f_{cm,theo}}$, were calculated and reported in Table 4. In addition, the mean absolute percentage error (MAPE), according to Equation (9), and the linear correlation index R$^2$ between the experimental and theoretical data were provided.

$$MAPE = \frac{100\%}{j} \sum \left| \frac{f_{cm,exp} - f_{cm,theo}}{f_{cm,exp}} \right| \tag{9}$$

where j is the number of considered data. A graphical representation of the results is illustrated in Figure 8 in order to make the comparison clearer. The green line (x = y) defines the perfect prediction domain, while the dashed lines individuate the 25%-scatter area. By considering continuous confinement, the Cascardi et al. model and the CNR DT215 model resulted in a comparable linear correlation (R$^2$ = 96%) and accuracy (mean $\frac{f_{cm,exp}}{f_{cm,theo}}$ equal to 0.81 and 1.33, respectively), while a different precision could be observed (Co.V = 10%–22%). In other words, the CNR-code furnished a similar scatter with respect to the experimental outcomes, irrespective of the number of layers, while for the other model [26], the same mentioned scatter increased with the number of layers. Referring to the discontinuous configuration, the scatter with respect to the experimental data remained lower than 25%. The mean value of $\frac{f_{cm,exp}}{f_{cm,theo}}$ was equal to 0.73 and 1.10 for the Cascardi et al. [26] model and the CNR DT215 [29] model, respectively.

The linear correlation index R$^2$ was found to be equal to 1 for both models and the COV values were 4% and 10%. Thus, for discontinuous configurations, the predictions appear less variable with the number of layers. Generally, the model in [29] lies on the conservative side while in [26], it is on the unconservative one. The MAPE ranged between 27.8% and 34.1% in the case of predictions by the Cascardi et al. model, while a scatter of 9.4%–21.7% was met with the CNR DT215 as foreseen. Therefore, the CNR DT215 exhibited a lower scatter when compared with the experimental outcomes related to the partially confined columns, while a comparable error was met by the two models with respect to the fully confined related data.

**Table 4.** Evaluation of the analytical models.

| Analytical Model | Label | $f_{cm,exp}$ (MPa) | $f_{cm,theo}$ (MPa) | $\frac{f_{cm,exp}}{f_{cm,theo}}$ | Mean ± Co.V | Median | MAPE (%) | R$^2$ |
|---|---|---|---|---|---|---|---|---|
| Cascardi et al. (2017) [26] | C-C-P-1L | 10.65 | 10.78 | 0.99 | | | | |
| | C-C-P-2L | 13.60 | 16.88 | 0.81 | 0.81 ± 22% | 0.81 | 27.8 | 0.96 |
| | C-C-P-3L | 15.31 | 24.17 | 0.63 | | | | |
| | C-D-P-1L | 6.97 | 8.94 | 0.78 | | | | |
| | C-D-P-2L | 9.41 | 12.96 | 0.73 | 0.75 ± 4% | 0.73 | 34.1 | 1.00 |
| | C-D-P-3L | 13.02 | 17.77 | 0.73 | | | | |
| CNR DT 215 (2018) [29] | C-C-P-1L | 10.65 | 7.67 | 1.39 | | | | |
| | C-C-P-2L | 13.60 | 10.26 | 1.33 | 1.29 ± 10% | 1.33 | 21.7 | 0.96 |
| | C-C-P-3L | 15.31 | 11.85 | 1.15 | | | | |

| | | | | | | | |
|---|---|---|---|---|---|---|---|
| C-D-P-1L | 6.97 | 6.88 | 1.01 | | | | |
| C-D-P-2L | 9.41 | 8.59 | 1.10 | 1.11 ± 10% | 1.10 | 9.4 | 1.00 |
| C-D-P-3L | 13.02 | 9.64 | 1.22 | | | | |

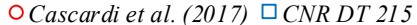

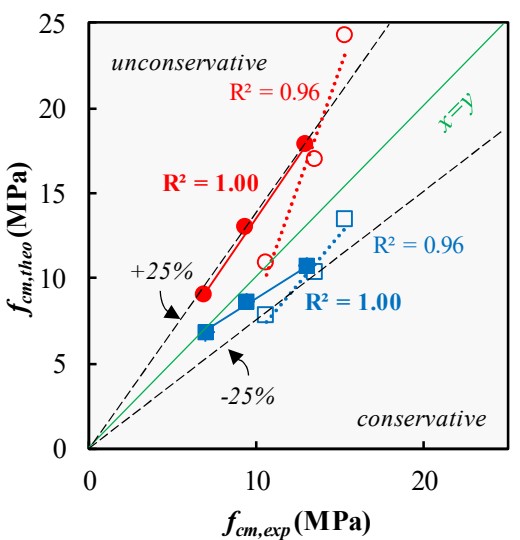

*solid marker = discontinuous confinement*

**Figure 8.** Experimental versus theoretical comparison of the axial strength.

In 2017, Cascardi et al. [27] performed an analysis-oriented model (AOM) able to predict the axial stress–strain curve for FRCM-confined columns, masonry, or concrete made by means of a step-by-step procedure. The matrix effect was considered by computing the k-step confining pressure according to the un-cracked or cracked behavior of the FRCM-system (see Figure 9). The proposed AOM was validated by comparison with the experimental curves as illustrated in Figure 10a–f. It was found that the predicted curve was generally stiffer in the first part, while a satisfactory agreement was noticeable for the post-peak slopes (Figure 10a–c).

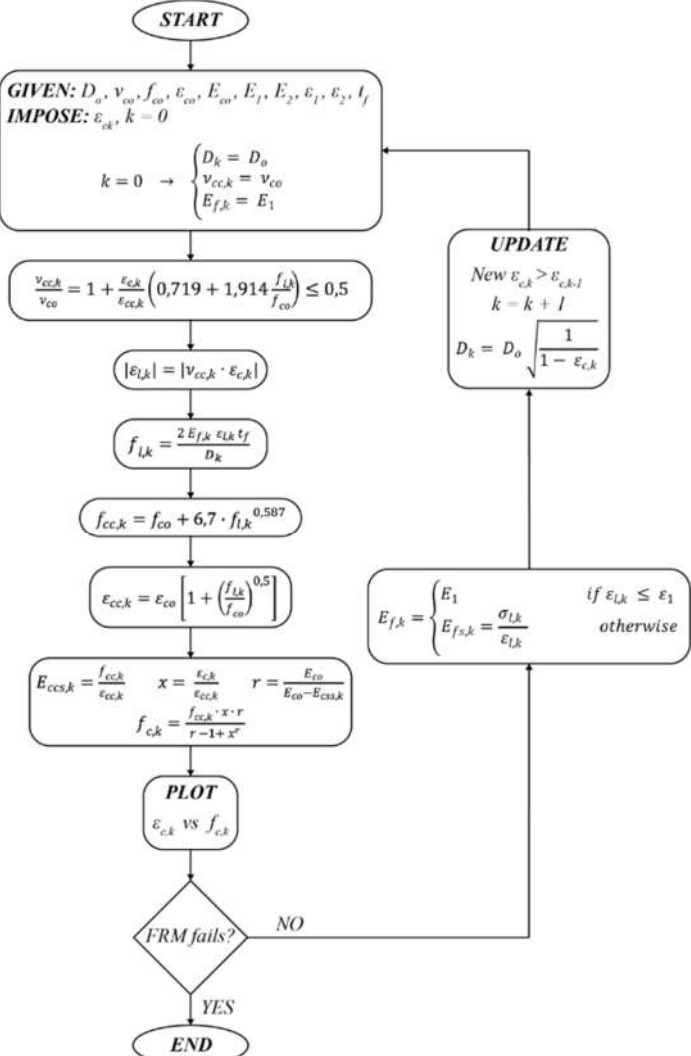

**Figure 9.** Flow-chart of the adopted analysis-oriented model (AOM). FRM, fabric reinforced mortar [27].

The AOM was also adapted to the case of partially-wrapping by implementing Equation (2) ([34]) in the procedure reported in Figure 9 and the forecast is illustrated in Figure 10d–f. The peak value was satisfactorily predicted, mostly referring to the discontinuous configuration, even if the shape of the whole curves appeared significantly different; this is because the theoretical outcome exhibited a softening (or plateau) post-peak trend while the experimental results were of the hardening type. The larger scatter between the analytical and theoretical curves was noticeable in terms of stiffness. Concerning the elastic initial stiffness, the analytical models assumed the compressive elastic modulus of the un-confined masonry in all cases, while the experimental findings showed a variation of the elastic initial stiffness (between 1443–3184 MPa) with the confinement configuration. Regarding the global trend, it should be stated that the experimental outcomes manifested a hardening behavior, while the used AOM was based on the assumption of softening post-peak trend, typically found for FRCM-confinement. The analyzed differences between the experimental and theoretical predictions suggest the need for further investigations aiming at improving the AOM model to include a wider variability of mechanical responses.

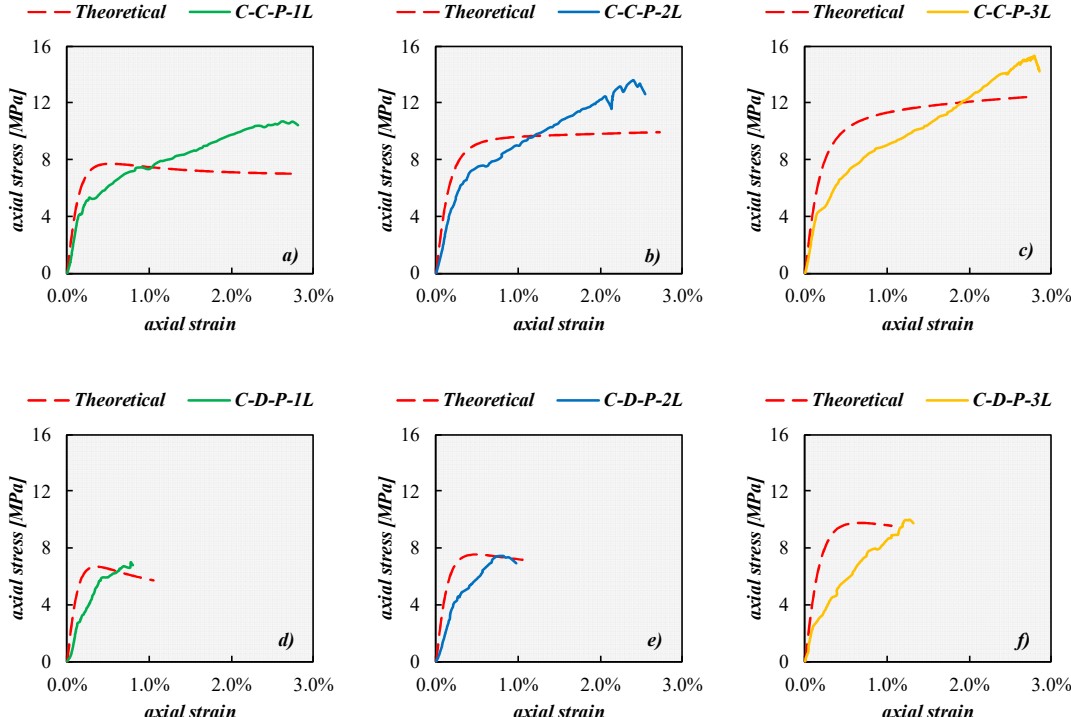

**Figure 10.** Experimental versus theoretical comparison of the axial stress–strain laws: (**a**) C-C-P-1L, (**b**) C-C-P-2L, (**c**) C-C-P-3L (**d**) C-D-P-1L, (**e**) C-D-P-2L, and (**f**) C-D-P-3L.

## 4. Conclusions

The paper described and discussed the results of an experimental and analytical investigation on the performances of clay brick masonry columns with a square cross-section confined by FRCM systems and subjected to an axial compression load. The obtained results allowed us to draw the following considerations:

- FRCM confinement systems improved the axial strength of the columns with respect to the unconfined ones in all cases;
- the confinement effectiveness increased with the number of FRCM-layers; in particular, passing from one to three layers, the axial strength gained about 50% and 43% for the continuous and discontinuous configuration, respectively. Moreover, the correlation between the number of layers and the compressive strength was linear for the case of the continuous-confined column (in fact 2-layer confinement resulted in 23% compressive strength gain) while the 2-layer partial confinement showed a 6% efficiency with respect to the single layer;
- partial-wrapping resulted in being less effective with respect to full-wrapping even if the load bearing capacity of the masonry core was improved;
- the theoretical prediction of the axial strength by means of available models was performed and the outcomes were found to be satisfactory enough for both continuous and discontinuous configurations (theoretical versus experimental scatter almost lesser than 25%);
- the theoretical prediction of the axial stress–strain behavior by means of available AOM was also performed and the curve matching were affected by a larger scatter, mostly in the post-peak branch; this result was due to the fact that a softening behavior was assumed by the model while a hardening result was found by the tests. For this reason, further experimental investigations are needed in order to better understand this aspect and perform possible improvements of the available AOM model.

**Author Contributions:** Conceptualization, L.O. and M.A.A.; Preparation of experimental investigation, S.V.; Analytical model, A.C.; Validation, L.O. and M.A.A.; Writing—original draft preparation, L.O. and S.V.; Writing—review and editing, A.C. and M.A.A.; Supervision, L.O. and M.A.A. All authors have read and agreed to the published version of the manuscript.

**Funding:** This research received no external funding.

**Acknowledgments:** The authors would like to express their appreciation to Ruregold s.r.l, Italy.

**Conflicts of Interest:** The authors declare no conflict of interest.

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
