# Peer review of "Confinement of Masonry Columns with the FRCM-System: Theoretical and Experimental Investigation"

_infrastructures, doi:10.3390/infrastructures5110101_

Round 1

Reviewer 1 Report

The paper presents an experimental study into the response of confined masonry columns with FRCM overlays. The main parameters investigated are the number of PBO layers and the continuity of confinement. Comparative assessment with existing analytical models is also undertaken. I had only a cursory look and the paper seems to be well written and organised and relatively clear throughout its length.

The paper needs some qualitative improvements by clearly highlighting the novelty of the study and its research significance. Also, although for a conference paper it would be acceptable, for a journal paper the literature review should cover papers outside of the authors’ research group.  It is strongly recommended that the literature review is expanded by referring to relevant literature.

There is some wording that would require changing (e.g. tensile stress to tensile strength, break deformation to ultimate or fracture deformation, depending at which displacement was this assessed) and clearer definitions of some parameters (e.g. ductility). Please have a close look at the terms and definitions and possibly use those available in Eurocodes.

The paper is expected to have an in-depth discussion of the results, rather than merely reporting the test results and fit of analytical functions. Could the authors add at least a couple of discussion paragraphs? For example, which are the implications of predicting a higher stiffness using the verified analytical models? What can be improved? Based on Fig 10, one can observe that the limit of proportionality and the stiffness of the two linear branches can be better predicted. This comparison can be carried out against existing literature and existing models for square RC columns where the confinement effects would be similar.

Author Response

Dear Reviewer,
the Authors thank you for your useful comments necessary for the improvement of the manuscript. Please find attached the answers to your questions.

Reviewer 2 Report

Please consider the following comments:

-Could you
include the results of the lateral displacement measurements?

-Looking at figure 7a, it appears that there is a noticeable improvement
between specimens C-C-P-1L and C-C-P-2L, but not between C-C-P-2L and C-C-P-3L (even the slope of the second branch in the stress-strain curve is the same). Although it is difficult to draw conclusions with such a small number of specimens, could this result indicate that there is a limit on the amount of reinforcement beyond which the effectiveness of the confinement is not increased?

-In my opinion, the last point of the conclusions is not supported by the results.

-There is a mistake in the number of the figure referenced in line 227.

Author Response

(The authors gave the same response as above.)

Reviewer 3 Report

Line 100 - Figure 2: sub-figures (a) and (b) should be exchanged to match the caption

Line 200 - Change reference "Table 3" to "Table 4".

Line 203 : change "the compression" to "the comparison"

Line 226: change "satisficing" to "satisfactory"

Line 248-248: a CoV "of less than 25%" cannot be described as accurate - please  rephrase so that the judgements in the conclusions more accurately describe the actual findings. 

Line 251: the term "accurate" is a rather exaggerated description of the matching of theoretical results to experimental measurements - the fit is not that good and the fact that the theoretical estimation of resistance is higher at early stages of deformation makes the estimation unconservative. 

Author Response

(The authors gave the same response as above.)

Round 2

Reviewer 1 Report

Thank you for addressing my comments and recommendations.